# Effectiveness of Ready-to-Use Therapeutic Food in Improving the Developmental Potential and Weight of Children Aged under Five with Severe Acute Malnourishment in Pakistan: A Pretest-Posttest Study

**DOI:** 10.3390/ijerph18179060

**Published:** 2021-08-27

**Authors:** Javeria Saleem, Rubeena Zakar, Gul Mehar Javaid Bukhari, Mahwish Naz, Faisal Mushtaq, Florian Fischer

**Affiliations:** 1Department of Public Health, University of the Punjab, Lahore 54590, Pakistan; javeria.hasan@hotmail.com (J.S.); rubeena499@gmail.com (R.Z.); 2Department of Community Medicine, Federal Medical and Dental College, Islamabad 44000, Pakistan; drgulmehar@gmail.com; 3Department of Primary and Secondary Health, Government of Punjab, Lahore 54000, Pakistan; dr.mahwish_naz@yahoo.com; 4Department of Biostatistics, Institute of Public Health, Lahore 54000, Pakistan; faisalskm1@gmail.com; 5Institute of Public Health, Charité—Universitätsmedizin Berlin, 10117 Berlin, Germany; 6Institute of Gerontological Health Services and Nursing Research, Ravensburg-Weingarten University of Applied Sciences, 88250 Weingarten, Germany

**Keywords:** development, nutrition, malnutrition, weight, children, ready-to-use therapeutic food

## Abstract

The objective of this study was to assess whether the standard therapy of ready-to-use therapeutic food in the treatment of uncomplicated severe acute malnutrition (SAM) is effective in improving developmental potential and weight gain in children aged under five years. A multicenter pretest-posttest study was conducted among 91 children aged under five with uncomplicated SAM in Pakistan. Study participants completed their eight weeks’ therapy of ready-to-use therapeutic food according to the World Health Organization’s (WHO) standard guidelines. The study outcome was the proportion of children with improved developmental potential in all domains in comparison with the pretreatment status and children gaining >15% of their baseline weight; mean weight-for-height/length z-score after completing eight weeks’ therapy of ready-to-use therapeutic food. The Denver Development Screening Tool II was used for developmental screening. Significant changes (*p* < 0.05) were observed for developmental status milestones in terms of gross motor, fine motor, and personal/social milestones, as well as language and global development milestones. There was a strong positive correlation (r = 0.961) between initial weight and weight at the last visit (*p* < 0.001). Ready-to-use therapeutic food is effective in improving development potential as well as promoting weight gain in children aged under five with uncomplicated SAM if provided according to WHO guidelines.

## 1. Introduction

Severe acute malnutrition (SAM), also known as severe wasting, is a highly prevalent and critical form of under-nutrition. Approximately 16.4 million children suffer from this condition globally [1]. Of these, approximately 1.4 million live in Pakistan, showing a case fatality rate of 6.2% [2]. The disease is characterized by muscle and fat loss, combined with increased systemic inflammation and susceptibility to infections [3]. SAM is categorized as complicated (20%) or uncomplicated (80%) depending on the diagnosis of medical complications (i.e., hypoglycemia, hypothermia, severe dehydration, acute infection of the lower respiratory tract, severe anemia, severe edema, high-grade fever, or loss of appetite) [4].

According to the standards of the World Health Organization (WHO), anthropometric criteria for the diagnosis of SAM are weight-for-height < −3 standard deviations (SD) or mid-upper arm circumference (MUAC) < 11.5 cm [4]. Furthermore, WHO guidelines mention that only those children who are suffering from malnutrition complications should be admitted to hospital; other children with SAM without complications (i.e., clinically well and alert, with good appetite) should be treated in the community setting in their homes [3,4,5].

Child development designates the progression of the child in all domains of human functioning, i.e., social, cognitive, motor, hearing, and speech [6]. Children surviving an acute episode of SAM are at high risk of suffering from long-lasting detrimental consequences to their physical health and development [7,8]. Adverse impacts have been observed on educational achievements, affecting economic productivity as adults [8,9,10]. Thus, timely recognition and management of severe wasting are needed. Nutritional interventions may increase children’s chances of survival, ensure a strong immune system, and improve their growth and development [5]. To deal with the SAM problem, a public health strategy referred to as the community-based management of severe acute malnutrition (CMAM), was launched in 2001. It is based on the early detection and treatment of children outside the facility in a local community setting [11,12]. Initially, this approach was known as the community-based therapeutic care model (CTC). It has been proven that, in low resource areas, the CTC approach is very useful and shows good results in comparison with facility-based treatments, because of its cost-effectiveness and wider population coverage [13,14]. 

Ready-to-use therapeutic food (RUTF) and basic medical care are the mainstays of treatment in a community setting for the management of uncomplicated SAM. RUTF is a high-calorie, energy-dense, micronutrient-enriched food [5,12,15]. It is a peanut-based paste wrapped in foil, with a weight of approximately 92 g and around 5.5 kcal/g, containing a high amount of protein and energy. It includes a balance of all necessary macronutrients and micronutrients [11]. Children aged 6–59 months can consume RUTF as a treatment for six to eight weeks according to their body weight, as recommended by the WHO [16,17]. 

The paradigm of the hospital-based approach to home treatment has merged with the effectiveness of RUTF to increase the acceptability of CMAM programs in the treatment of children with SAM in a cost-effective manner [18]. The Pakistani government has endorsed the CMAM approach to treating severely acutely malnourished children. The national guidelines were first developed in 2010, followed by updates in 2015 after the emergence of new evidence [19]. Previous research in Ghana, Ethiopia, and Pakistan has proven the effectiveness of this community-based intervention with RUTF in improving the weight of children [13,14,20].

To our knowledge, all of these studies or retrospective study designs only look for children’s weight gain. No study has been conducted with a prospective design to demonstrate that this micronutrient-enriched paste is also effective in improving the development quotient of children along with their weight. For that reason, we hypothesized that ready-to-use therapeutic food would be equally effective in improving developmental potential along with weight gain in the management of children with SAM. We tested this hypothesis by conducting a pretest-posttest study in rural areas of Pakistan to check the efficacy of RUTF for child development along with growth.

## 2. Materials and Methods

### 2.1. Study Design and Setting

A multicenter pretest-posttest study was conducted at three basic health units and one rural health center in Dera Ghazi Khan, a district of Southern Punjab, Pakistan. This district has a high illiteracy rate and the majority of the population consists of individuals with low socio-economic status. It is also an underprivileged district that has a high prevalence of malnutrition and poverty [21]. Children aged 6 to 59 months and suffering from SAM were recruited before starting their nutritional treatment at the outpatient care centers of these health units. 

Overall, 91 children, both girls and boys, were enrolled after their parents gave permission. The inclusion criterion was severe wasting accessed by the WHO protocols (weight for height < −3 SD and MUAC < 11.5 cm), which was not accompanied by any complications of malnutrition [3,4]. Children with physical or learning disabilities, or who were clinically unfit, were excluded from the study. Children tested as being clinically unwell were not included in the study and were instead referred to the hospital for complete clinical evaluation.

The sample size was calculated by the following formula, keeping the power of the study equal to 90% and the level of significance equal to 5%:(1)$$n=\frac{2\sigma^{2}{\left(z_{1-\alpha /2}+z_{1-\beta }\right)}^{2}}{{\left(\mu_{1}-\mu_{2}\right)}^{2}}$$

Desired power = 90%

Desired level of significance = 5%

Population standard variance = 12.8881 [22]

Population mean weight before = 5.08 kg

Population mean weight gain = 3.3 kg

According to this formula, a sample size of 86 participants was required for the analyses. We decided to recruit 100 children to allow for non-responses in the posttest.

### 2.2. Baseline (Pretest) Assessment

A pretested sociodemographic and nutritional questionnaire was utilized to obtain information about the child’s age, gender, household income, household size, immunization status, parents’ education, parents’ professions, history of infections, complimentary feeding practices, breastfeeding, and access to medical assistance. Family monthly income from all sources was asked for in Pakistani rupees, and children’s current immunization status was checked from their immunization card according to the Pakistan Expanded Program on Immunization [23]. In this regard, “complete” indicates that the child had received all age-appropriate immunizations, whereas “incomplete” indicates that he or she had not received all age-appropriate immunizations. Any repeated episode of illness in the past six months due to diarrhea, respiratory tract infections, or any other morbidity were obtained from the children’s hospital reports and healthcare prescriptions. A history of exclusive breastfeeding (defined as receiving only mother’s milk up to the age of six months) and complimentary feeding practices (food quantity, variety, and frequency) by the mothers were evaluated in light of WHO recommendations on infant and young child feeding practices [24]. Maternal hygiene practices were judged by inquiring about their own and children’s handwashing practices before eating food and after using the toilet. The mother’s habits were further assessed by asking about cleaning and hygienic preparation of food before cooking, with proper food storage.

A child appetite test was performed by offering a small sample of RUTF and watching how much the child consumed of the offered amount. After three feeding attempts, a child was labelled as having poor appetite if they failed to eat a minimum of one third of a pack of RUTF. As per the protocol, these children were also not included in this study and were transferred to hospital.

A physical examination according to WHO and national guidelines was then carried out to assess the children’s alertness, hydration status, severe palmar paleness, severe edema, lower respiratory tract infections, hyperpyrexia, and hypoglycemia before enrolment and start of treatment [3,19]. Where such complications existed, the children were also not registered for this study and were shifted to hospital for intensive treatment.

In cases of facility-based delivery, the children’s gestational age was obtained by consulting the antenatal record; for home delivery, information was supported by maternal report. For children who were ≤ 24 months of age and born prematurely, before 37 weeks of gestation, their age was adjusted by deducting the total weeks of missed gestation from the present age. This information was acquired from mothers and caregivers at health units.

### 2.3. Anthropometric Assessments

The anthropometric evaluations were carried out by qualified nutrition supervisors who were specially trained for these evaluations. Weight was measured using the UNISCALE to the nearest 10 g by weighing children in very light clothing or, if required, without clothing. When a child was unable to stand, their mother was weighed while holding the child and afterwards the mother’s weight was subtracted. The child’s length was estimated using a length measuring board (SECA GmbH & Co. KG, Hamburg, Germany) to around 0.1 cm accuracy. The height of children who were able to stand, and who were more than 87 cm tall, was measured in the standing position without shoes [3,4]. Weight-for-height z-scores were evaluated in accordance with the WHO child growth standards using “WHO ANTHRO, version 3.2.2”.

### 2.4. Developmental Assessment

Once the eligibility criteria were met, children undertook a developmental assessment with the support of a pediatrician by following the Denver II Developmental Screening Tool (DDST II) [25]. This developmental tool evaluates a child’s capacity up to the age of six years to complete a variety of different tasks and then compares the child with a standard population of children of similar age. Tasks are organized into four areas: Personal social development, fine motor milestones, language proficiency, and gross motor milestones. Following a standardized procedure, the assessment was assessed in each domain as “normal” (if the child is successful in completing the designated tasks that fall to the right of the age line), “caution” as an intermediate category (if the child is not able to perform tasks on which the age line falls on or is between the 75th and 90th percentiles), or “delayed” (if the child fails to perform a task that 90% of a standard population of children of similar age performed at an earlier age or when a child fails or refuses an item that falls completely to the left of the age line). Based on these domains, the final global developmental status of children was completed and scored as “normal” (no category delayed and no more than one category classified as caution), “suspect” (≥2 cautions or ≥1 delay) or “untestable” (if development assessment was not completed due to child refusal) [25]. 

The study outcome was the proportion of children with improved developmental potential in all domains in comparison with the pre-treatment developmental potential, and children gaining > 15% of their baseline weight, mean weight-for-height/length z-score, after completing a standard eight weeks’ therapy of ready-to-use therapeutic food.

### 2.5. Intervention

All of the children were enrolled for eight weeks’ therapy of RUTF provided by the Department of Health of the Government of Punjab, sponsored by UNICEF, at CMAM centers. By following national and WHO guidelines, RUTF was provided to families weekly according to the weight of the children, which equates to approximately 1.5–5.0 sachets/day [3,19]. RUTF was given to parents by well-trained staff in CMAM centers. The staff also demonstrated to parents how the RUTF should be given and informed them about the benefits of RUTF.

### 2.6. Follow-Up Assessments

Parents were provided with a CMAM registration card and were assisted to attend outpatient centers with their children. During the study’s duration, and in line with standard practice, children were examined on a weekly basis at CMAM centers. The reason for this was to acquire their RUTF and be assessed for any medical or nutritional complication. In cases where complications were reported, children were referred to hospital for further evaluation and treatment. Post-treatment anthropometry and developmental assessment of the children were conducted by CMAM staff and a study pediatrician after they had completed their eight-week course of RUTF.

### 2.7. Statistical Analysis

Data was entered and cleaned using SPSS version 25.0. Frequencies and percentages of categorical variables like age, gender, monthly income, and educational status were calculated. Mean (SD) was calculated for the pre- and post-intervention quantitative variables, such as weight and weight for length/height z scores. The Chi-Square test was used to assess the association between pre- and post-intervention developmental status. A paired *t*-test was used to compare paired observations for z-scores and weight gain. A *p*-value of <0.05 was taken as statistically significant.

### 2.8. Ethical Considerations

Written informed consent was received from caregivers or parents after the study had been explained to them. Ethical approval for the current study was obtained from the District Health Office of Dera Ghazi Khan (Ref.: 838/ADGHS/IRMNCH) and the Ethical Review Board of the University of the Punjab.

## 3. Results

Overall, 100 children with uncomplicated SAM were enrolled in the study; 91 of them completed the study in line with the study protocol and were included in the analysis. The mean age was 15.2 ± 9.69 months, and most children were aged < 12 months (60.4%). More than half of study participants were females (58.2%) and the majority had a low household income. The mean height was 66.68 ± 8.97 cm, mid-upper arm circumference was 9.89 ± 0.99 cm. Immunization was mostly completed (79.1%), and most (81.3%) were not exclusively breastfed. Poor hygiene practices were witnessed among most participants (93.4%). A weight gain of >15% in the eight weeks of eating RUTF was observed in most cases (86.8%) (Table 1). 

Table 2 shows that significant changes between pretest and posttest were observed for global development and all developmental status milestones. The largest increase is visible for global development. The increase was not so large for fine motor development. However, this developmental status milestone already showed good results at the pretest. Table 3 shows the baseline and follow-up characteristics of weight and weight-for-height z-score (5.09 ± 1.54 kg vs. 6.48 ± 1.58 kg; −4.07 ± 1.27 vs. −1.25 ± 1.99). It indicates significant positive effects on the attributes; males and females responded equally well to the therapy.

## 4. Discussion

This study indicates that RUTF is not only effective in encouraging weight gain, but also in improving developmental potential among children aged under five. Many studies have already been conducted on the effectiveness of the CMAM program in the community setting. However, until now only limited evidence has been available on the effectiveness of RUTF in improving the development quotient of children along with weight gain.

Previous research has concluded that there is an inverse relationship between child development and nutritional status, because children with low nutritional status are unable to attain their full developmental potential in order to lead a productive life [7,8,26]. Our study results were consistent with these studies, as the children showed delayed development in all domains in pretreatment developmental screening.

Inadequate nutrient supply may cause abnormalities such as reduced brain size, adverse impacts on cell maturation resulting in behavioral abnormalities, and delay in personal social development [7,27]. In malnourished children, deficiencies in both macronutrients and micronutrients are the main causes of delayed development, because nutrients such as iron, calcium, zinc, vitamins A and D, and folate are vital for cognitive functioning, along with reaching motor and language milestones [7,27,28]. During the first two years of life, the risk of developmental delay is much higher. There is a possibility that an infant’s risk of developmental delay is increased if he or she is malnourished during early infancy. Additionally, it is a sign of major health or psychosocial issues. Infancy and toddlerhood are periods of rapid development that are easily influenced by socio-demographic and environmental factors [7,27,28].

For treating children with uncomplicated SAM, effective community-based care for acute malnutrition, including RUTF, has established a solid reputation. The developmental delay was significantly reduced with the treatment of uncomplicated SAM using RUTF in terms of the gross motor, personal/social, fine motor, language and global development assessed through Denver II. The study results were consistent with a study from Pakistan and Ethiopia qualitatively assessing the barriers to SAM treatment [29].

After an eight-week follow up with RUTF management, we found that children with uncomplicated SAM showed profound improvement in all developmental domains in comparison with their pretreatment developmental potential. RUTF contains therapeutic doses of all the micronutrients and macronutrients required for the catch-up growth and brain development of children [11,12]. Thus, with timely recognition and management, it is evident that children can attain their full developmental potential, but delays in treatment may lead to irreversible damage [5,7,28]. 

According to WHO guidelines, the proper utilization of recommended RUTF is key for attaining the required weight gain and complete recovery [13]. We have also found that, after an eight-week treatment with RUTF, children achieve immense gains in mean weight and mean weight-for-height z-scores. These findings are in line with previous studies conducted in Ethiopia, Ghana, and Pakistan, where children achieve their required weight after treatment with RUTF [13,14,20,28]. 

The strength of our research is that developmental screening was conducted (1) by pediatricians who were experts in the developmental testing of children and (2) by using validated and reliable development screening scales [25]. The rate of loss to follow-up and the defaulter rate are low in our study and in line with references from the Sphere Project [30], because we have properly followed each child in the community and with the help of female health visitors in the areas to make sure that the children adhered to RUTF eating regimes and follow-up visits.

The limitations of our study relate to the comparatively short duration of the study period. Because of this short duration, we could not determine whether children were able to maintain this recovery of weight and development in order to gain the maximum advantage in the long term. Another limitation was that, due to financial constraints, we could not conduct any lab tests to assess the deficiency and recovery of micronutrient status in the pre- and post-examination phases. The lack of a control group was also a limitation, meaning that true differences could not be attributed as similar findings might have been noted in the control group. Further studies with an increased sample size, duration, and effect of therapies other than RUTF, along with biochemical testing in diverse locations, are required in order to advance and replicate these promising results. Furthermore, a randomized controlled study design with a control group is needed.

## 5. Conclusions

The study concluded that RUTF is beneficial for improving the developmental potential and weight gain of children with SAM aged under five if it is given in a timely manner and in line with the WHO recommended amounts. Community-based programs should be implemented and supported in order to gain improvements at the population level, particularly for vulnerable subgroups.

## Figures and Tables

**Table 1 ijerph-18-09060-t001:** Characteristics of the sample (*n* = 91).

Variables	Categories	*n* (%)
Age	<12 months	55 (60.4)
	12–24 months	24 (26.4)
	>25 months	12 (13.2)
Sex of child	Males	38 (41.8)
	Females	53 (58.2)
Monthly income (in PKR)	15,000–35,000	35 (38.5)
	<15,000	56 (61.5)
Father’s education	No education	51 (56.0)
	Primary and above	40 (44.0)
Mother’s education	No education	61 (67.0)
	Primary and above	30 (33.0)
Immunization	Incomplete	19 (20.9)
	Complete	72 (79.1)
Episodes of illness in last six months	≤7	42 (46.2)
	>7	49 (53.8)
Exclusive breastfeeding	Yes	17 (18.7)
	No	74 (81.3)
Complimentary feeding practices	Appropriate	21 (23.1)
	Inappropriate	70 (76.9)
Hygiene practices	Good	6 (6.6)
	Poor	85 (93.4)
Under-five siblings	0–2	72 (79.1)
	3 and more	19 (20.9)
Weight gain on RUTF in eight weeks	≤15%	12 (13.2)
	>15%	79 (86.8)

PKR: Pakistan Rupees; RUTF: Ready-to-Use Therapeutic Food.

**Table 2 ijerph-18-09060-t002:** Developmental status at pretest and posttest (*n* = 91).

Milestones	Status of Milestone	Pretest		Posttest		*p*-Value ^1^
		*n*	%	*n*	%	
Gross motor	Normal	58	63.7	73	80.2	<0.001
	Suspect	33	36.3	18	19.8	
Fine motor	Normal	54	59.3	63	69.2	0.022
	Suspect	37	40.7	28	30.8	
Personal/social	Normal	34	37.4	50	54.9	<0.001
	Suspect	57	62.6	41	45.1	
Language	Normal	62	68.1	72	79.1	0.002
	Suspect	29	31.9	19	20.9	
Global Development	Normal	33	36.3	54	60.0	<0.001
	Suspect	58	63.7	36	40.0	

^1^*p*-value calculated using Chi-Square test.

**Table 3 ijerph-18-09060-t003:** Pretest and posttest weight comparisons among children receiving RUTF (*n* = 91).

**Variables**	**Number of Visit**	**Mean (SD)**
Weight-for-length/height z-score	First Visit	−4.07 (1.27)
	Last Visit	−1.25 (1.99)
Weight (kg)	First Visit	5.09 (1.54)
	Last Visit	6.48 (1.58)
**Variables**	**Paired *t*-test**	***p*-value**
Weight-for-length/height z-score at first visit with z-score weight-for-height at last visit	−16.3	<0.001
Weight at first visit with weight at last visit	−30.96	<0.001

## Data Availability

Data is available from corresponding author upon reasonable request.

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
