# Peer review of "Effectiveness of Ready-to-Use Therapeutic Food in Improving the Developmental Potential and Weight of Children Aged under Five with Severe Acute Malnourishment in Pakistan: A Pretest-Posttest Study"

_ijerph, 2021, doi:10.3390/ijerph18179060_

Round 1

Reviewer 1 Report

This is a very interesting study, on a sensitive topic. However, there are several points that need to be improved

1) Introduction:  the introduction of the manuscript is lengthy, and does not fully support the aim of the study. For instance, the developmental milestones are only referred in the last paragraph, when the authors suggest that such an examination is relatively novel

2) Methods: Apart some English editing that is required, the methods section is clear and concise

3) Results: The authors need to report z-scores for weight and height beyond the raw anthropoemtric values. This should be done both in the baseline and follow up assessment

I do not understand Table 3. What does the column Paire-t test stand for? I would suggest you present us the mean weight/height z-score gain

Figure 1 is also confusing. This strong correlation suggests that the higher the initial weight, the gretaer the weight gain with RUTF. Why is this of importance? For instance, could this mean that children with uncomplicated SAM, there is a cutoff for weight-for-age that children may be more benefited from RUTF?

4) Discussion: The discussion section needs major modification. The authors do not clearly emphasize on the strengths of teir analyses

The lack of a control group is also a major limitation of the study design, which is not discussed. Moreover, given you performed poewer analysis, and your sample is adequate, I do not understand why you discuss that your sample is small.

The added value of the developmental outcomes that were examined in this study is not adequately discussed

Reviewer 2 Report

Overall I believe this project was a great idea. Measuring the impact of RUTF prospectively on growth and development is an important topic. The authors provided good details on the methods which was commendable. I think the results can be better described by giving a little more description of the contents placed on the tables especially table 3. Also figure I needs a little more description. The discussion can also be improved a little more by discussing more on nutrition and growth and development. Also, authors need to  include some information on other factors that could have influenced improvements in outcomes since there was no control group.

I think the main concern I had was the language. In some sentences, connecting words are missing and some sentences are incomplete. I highlighted just a few on the attached document. In some places, tenses are mixed making the content hard to comprehend. I believe the paper should read in past tense. There are some sentences where words such as "has been" was used instead of "was".

Author Response

Please find the reply in the document attached.

Reviewer 3 Report

This study aimed to assess the effects of ready-to-use therapeutic food on the developmental potential and weight gain of children under five years with uncomplicated severe acute malnutrition. Some comments are listed below:

1) Introduction: I suggest to simplify this part and emphasize the innovations of the study.

2) Methods: How did the quality control performed in the process of developmental assessment?

3) Results: What was the purpose for the correlation analysis between initial and final measures of weight/Z scores? I suggest to delete Figure 1.

4) Results: Did the boys and girls respond equally well to the therapy?

5) Results: the mean(SD) of weight/Z score before and after therapy should be listed.

6) Results: SAM is categorized as complicated (20%) or uncomplicated (80%). What types of SAM for the children enrolled in the presented study? What the main reasons for SAM of these children?

7) Results: How can the improvement of the developmental status be attributed to therapy rather than other factors influencing the development of children (e.g age)?

8) Discussion: Except for RTUF, are there other therapies for SAM? What about the effects of other therapies in comparison with RTUF?

Author Response

(The authors gave the same response as above.)

Round 2

Reviewer 3 Report

The authors have answered all of my questions